# Wound Pain as a Determinant of Function in Patients Hospitalised for Burns

**DOI:** 10.3390/ijerph20031951

**Published:** 2023-01-20

**Authors:** Anna Budzyńska, Dorota Talarska, Grażyna Bączyk

**Affiliations:** 1Department of Nursing Practice, Poznan University of Medical Sciences, 61-701 Poznan, Poland; 2Department of Preventive Medicine, Poznan University of Medical Sciences, 61-701 Poznan, Poland

**Keywords:** burn, depression, pain, functioning, satisfaction with appearance

## Abstract

Burn wounds most often occur on visible parts of the body. They therefore cause fear of disfigurement and rejection by those around them. In addition, they cause pain. The main aim of this study was to analyse factors affecting the functioning of patients hospitalised for burns. The study included 57 patients hospitalised for burns. Each person was surveyed twice: on day seven after admission and on the day of discharge from the hospital. The following research tools were used: a personal questionnaire to collect clinical data and the scales of the Katz Activities of Daily Living (ADL), the short-form McGill Pain Questionnaire (SF-MPQ), the Beck Depression Inventory (BDI) and the Satisfaction with Appearance Scale (SWAP). On the discharge day, compared to day 7, there was an improvement in the patient’s level of functioning in all areas analysed. Pain intensity (*p* < 0.0001), depression (*p* < 0.0001) and dissatisfaction with appearance (*p* = 0.0239) decreased significantly. Improvements in daily functioning were also obtained (*p* < 0.0001). Multiple regression was performed to estimate the effect of selected variables on patients’ level of functioning. Burn area (*p* = 0.0126), pain level (questionnaire part B: *p* = 0.0004 and questionnaire part C: *p* = 0.0208) and gender (*p* = 0.0413) had a significant impact on the level of daily functioning. Pain complaints are the most crucial predictor affecting the functional status and depression levels in burn wound patients. Increased levels of depression promote dissatisfaction with one’s appearance.

## 1. Introduction

The way people function depends on many factors. The most frequently mentioned are state of health, limitations due to disability and increased symptoms of illness, mood, depression and financial situation. In addition, in the psychosocial sphere, self-esteem has a significant influence. How one perceives oneself, including one’s appearance both by oneself and by those around one, is essential in the construction of one’s own identity [1,2]. People with visible changes in appearance, such as body deformities, scars and limb amputations, often fear a lack of acceptance from society, especially from the immediate environment, i.e., life partners, friends and the place of work or study. Therefore, the immediate environment’s attitude impacts their psychophysical functioning and life and career plans [3,4]. The first people those with burn wounds come into contact with are healthcare professionals. Their behaviour, especially the degree of interest in the patient’s problems, affects their mood and level of involvement in the healing process. A burn is a traumatic event affecting the psychophysical functioning of patients [1]. An additional negative factor directly related to extensive or deep burns is the long period of hospitalisation, changes in appearance, pain and the incurring of social and material costs for the injured due to the suspension or inability to pursue professional activities [5,6,7,8]. Lack of adequate support during the initial period of hospitalisation can cause a significant lowering of mood, feelings of isolation and low self-esteem, exacerbating pain and anxiety [5,9,10]. Later, depression and fear of living in the community can be experienced [11].

During hospitalisation, the focus is on wound healing and life-saving measures. At the same time, the patient’s emotional needs are often overlooked or not given sufficient attention [5,12]. Furthermore, psychological functioning, including mood, degree of self-acceptance and appearance and adaptation to hospital conditions, translates into physical functioning, including the ability to undertake activities of daily living [11].

It is important to involve patients as early as possible in basic activities of daily living (for example, washing, eating and dressing). Their increased participation in self-care activities leads to increased self-esteem and independence [13].

The most common psychological problems faced by patients with burn wounds include depression, anxiety, post-traumatic stress disorder, fear of disfigurement, social isolation, pain and the financial burden that follows the lengthy hospitalisation and necessary treatment [7].

It is worth noting that there are few research papers on patient functioning during hospitalisation. Reports mainly concern the post-hospitalisation period and show how patients with a burn scar cope psychologically, physically and socially after such a harrowing experience. Knowledge paucity on how a burn patient functions during hospitalisation has revealed the need to learn more about the problems in this group of patients.

The main aim of this study was to analyse the factors affecting the functioning of patients hospitalised for burns. In addition, the aim was to compare patients’ functioning on day seven after admission to the burn unit and the day of discharge.

## 2. Materials and Methods

The study included 57 patients hospitalised for burns at the Burns Treatment Unit of the Department of Trauma, Burns Treatment and Plastic Surgery of the Poznan University of Medical Sciences at the Józef Strusia Multi-Specialist City Hospital in Poznan between September 2018 and February 2020.

All scales and the interview questionnaire were aimed at adults who gave informed consent to participate in the study. Before the study, the purpose was explained, and patients were informed that their answers would remain anonymous. All questionnaires were completed in the presence of one nurse observer.

Patients hospitalised for burns and whose hospital treatment lasted a minimum of 14 days were included in the study. The following patients were excluded from the study: those with current and past psychiatric disorders, alcohol dependence syndrome, cancer and those in a pharmacological coma. Each person was examined twice: on day seven after admission and on the day of discharge from the hospital.

In the first stage of the study, a personal questionnaire was used (sociodemographic data were collected), and clinical information was collected (data from medical records). In addition, the scales of the Katz Activities of Daily Living (ADL) scales, the short-form McGill Pain Questionnaire (SF-MPQ), the Beck Depression Inventory (BDI) and the Satisfaction With Appearance Scale (SWAP) were used. In the second stage, clinical information in a personal questionnaire was completed, and the scales from the first stage were used again.

Wallace’s rule of nines was used to determine the burn area of the body: total body surface area (TBSA).

Figure 1 describes the study.

In conjunction with the aim of the study, variables were identified, and appropriate research tools were selected. The dependent variable was the functioning of patients with a burn wound: daily functioning—ADL scale, mental functioning—Beck Depression scale.

Independent variable—sociodemographic characteristics, wound characteristics, pain intensity—the short-form McGill Pain Questionnaire (SF-MPQ) and satisfaction with one’s appearance—Satisfaction With Appearance Scale (SWAP).

### Questionnaires

A survey to collect sociodemographic and clinical data.The Katz Activities of Daily Living (ADL) Scale—assesses the performance of basic activities such as getting out of bed and moving to a chair, dressing and undressing, bathing, using the toilet, eating independently and controlling bowel movements. The patient can earn 1 point for each activity performed independently. The maximum is 6 points. The following criteria are distinguished: able-bodied 5–6 points; moderately able-bodied 3–4 points; severely disabled 2–0 points [14,15]. All activities assessed by the ADL scale were defined as daily functioning, and this term was used in further analyses.The Polish version of the short-form McGill Pain Questionnaire (SF-MPQ)—provides a qualitative and quantitative measurement of pain sensations. It consists of three parts. The first refers to 15 adjectives describing pain experienced during the past week, including 11 sensory and four affective pain items. The patient rates each item on the following pain intensity scale: 0—no pain, 1—mild pain, 2—moderate pain and 3—severe pain. A maximum of 45 points can be given to the patient (SF-MPQ—part A) [16,17,18]. The second part includes the Visual Analogue Scale (VAS), which assesses pain experienced during the last week. On a graphical 10 cm section, the patient marks the severity of the pain experienced. This 10 cm line represents the pain of increasing severity, from ‘no pain’ to ‘worst pain possible’. The final score is in millimetres (SF-MPQ—part B) [16,19]. The final part indicates the severity of the pain experienced at the time of the test (Present Pain Intensity, PPI). The following categories are distinguished: 0—no pain, 1—mild pain, 2—moderate pain, 3—severe pain, 4—very severe pain, 5—unbearable pain (SF-MPQ—part C) [16,17,20].The Beck Depression Inventory (BDI)—Polish version—a screening tool. Used to assess the presence of depressive symptoms resulting from adjustment disorders. This scale contains 21 items. Each item consists of four statements describing how the patient felt during the last day of the past week or month. The subject selects one of the four affirmative statements and scores on a scale from 0 to 3 points. The total score is the sum of the points obtained from the 21 statements [21,22]. There are the following ranges: 0–11 points—no depression; 12–26 points—mild depression; 27–49 points—moderately severe depression; 50–63 points-very severe depression [22].Satisfaction With Appearance Scale (SWAP)—the scale contains 14 items relating to patients’ thoughts and feelings about their appearance following a burn. Each of these items is scored by the patient in a range of 0 points (strongly disagree) to 6 points (strongly agree). The total score is the sum of all the points scored. Questions 4–11 are scored inversely. The higher the score, the greater the dissatisfaction with one’s appearance and body image. A maximum of 84 points can be scored [23,24,25]. The SWAP scale contains the following four areas:
Subjective satisfaction with appearance, highlighting facial features (elements 4–7);Subjective satisfaction with appearance, emphasising other parts of the body (elements 8–11);Social discomfort due to appearance (elements 1–3);Social impact related to appearance (elements 12–14) [25,26].

## 3. Statistical Analysis

Count (N) and percentage (%) present statistical descriptions for qualitative variables. The mean and standard deviation (SD) present quantitative variables. For these variables, the conformity of their distribution to the normal distribution was checked using the Shapiro–Wilk test. This test did not confirm consistency with the Gaussian curve for most variables, so in this paper, all analyses were performed using non-parametric tests. The non-parametric Wilcoxon test was used to compare the results of the level of functioning obtained on day seven of hospitalisation and those obtained on the day of hospital discharge. A comparative analysis of the level of functioning according to gender, place of residence, occupational status, burn area and occurrence of burn wound infection was performed using the non-parametric Mann–Whitney *U* test.

On the other hand, the Kruskal–Wallis test was used for comparative analysis of the level of functioning according to education, marital status, degree of burn and location of the scar. Further analyses were performed with multiple comparison tests (post hoc tests) if the test detected any differences. Relationships between selected variables were analysed using the non-parametric Spearman rank correlation test.

In this study, multivariate regression analysis was performed to test the effect of selected independent variables on psychophysical variables. In addition, descriptive statistics and comparative analysis were performed using the statistical programme STATISTICA, version 13 (StatSoft, Inc., Tulsa, OK, USA). Results were considered significant if *p* < 0.05.

## 4. Characteristics of the Studied Group

The mean age of the entire study group was 48.5 ± 13.7 years (Table 1). The predominant group was male (n = 40), with a mean age of 46.8 ± 14.4 years. The mean age of women was higher at 52.7 ± 11.4 years. The most numerous respondents had a vocational education (49.1%). The fewest were those with primary education (7.0%). In total, 63.1% of the respondents were married. Most respondents lived in the city (64.9%) and were engaged in professional activities (73.7%). Patients with partial-thickness/full-thickness burns constituted the most numerous respondents (42.1%). An overwhelming number of patients had suffered a thermal burn (92.9%) and 40.4% of patients indicated that their hands had the worst possible burn scar (Table 1).

The most common areas of burn for 70.2% of the study participants were the head, face, hands and neck. Hospitalisation of patients for burns lasted an average of 28 ± 10 days. According to TBSA (total burn surface area), the average burn area was 16.56 ± 9.46%. Almost half of the patients (45.6%) took one antibiotic during their hospital stay. Symptoms of burn wound infection during the entire hospitalisation occurred in 18 patients, representing 31.6%.

## 5. Results

When analysing the patients’ level of independence in basic activities of daily living (ADL scale), it was found that on day seven of the hospital stay, there was a predominance of able-bodied (n = 27, 47%) and moderately non-disabled patients (n = 23, 40.5%). The mean ADL scale score was 4.39 points. (Table 2). On discharge from the hospital, 96% of the subjects fell into group one, i.e., the able-bodied group, with a mean ADL score of 5.84. This result means that the hospitalisation had a beneficial effect on the return to complete independence.

The independent variable included in the study was a pain in the wound area. On day seven, the mean score obtained in the pain intensity part (questionnaire part A) of the SF-MPQ scale was 17.51 points, in questionnaire part B—64.11 points and questionnaire part C—1.49 points. It decreased significantly on the day of discharge.

More than half (56%) of the subjects had a lowered mood corresponding to BDI scale level—mild depression on day 7 of hospitalisation, while 30% of the patients had a lowered mood on the day of hospital discharge (Table 2).

As assessed using the SWAP scale, satisfaction with appearance averaged 31.47 points on day 7 and 26.46 points on the day of discharge. Women were significantly less satisfied with their appearance than men in the first and second study periods. Only in men did satisfaction with appearance increase on the day of discharge compared to day 7 of hospitalisation.

Table 2 compares patients’ functioning on day seven of hospitalisation and the day of discharge. All analysed areas showed statistically significant improvement on the day of discharge.

The following sociodemographic factors influenced the level of daily functioning only in the first period of hospitalisation: gender (*p* = 0.0091)—men showed greater efficiency, and place of residence (*p* = 0.0236)—patients living in rural areas were more efficient than city dwellers. The depressive symptoms correlated only with gender but in both study periods (*p* = 0.0016 and *p* < 0.0001) more depressed mood occurred in the first period in men and on the day of discharge in women.

Of the clinical factors (degree of burn, burn area, burn area and wound infection), there was a correlation in the first period of hospitalisation between the level of daily functioning and burn area (*p* = 0.0126); ADL functioning worsened with increasing burn area. On the other hand, the number of hospitalisation days negatively correlated (Rs −0.29, *p* = 0.0267) with the level of daily functioning and positively with the severity of depressive symptoms. The rule was that as the number of days spent in the hospital increased, patients’ levels of self-efficacy decreased statistically significantly, and levels of depression increased (Rs 0.37, *p* = 0.0044).

A difference in the severity of depressive symptoms was in patients with and without wound infection (*p* = 0.0394). There were lower levels of depression among patients without infection symptoms. Our study outlined a statistically significant difference between appearance satisfaction depending on the degree of burn and the area of the burn in period one. In the case of burn severity, a significant difference between full-thickness and partial-thickness burns was evident. Patients with partial-thickness burns (*p* = 0.0335) were more dissatisfied with their appearance. When assessing the area of the burn, it was noted that on day seven of the study, patients were more dissatisfied with their appearance when burned in areas such as the head, face, hands or neck than the rest of the body (*p* = 0.0147).

The analysis also showed positive correlations between the level of pain assessed by the SF-MPQ scale and the SWAP and BDI scale scores. In addition, positive correlations were obtained between pain level (questionnaire parts A and B of the SF-MPQ scale) and satisfaction with appearance (A: Rs 0.33, *p* = 0.0113, B: Rs 0.42, *p* = 0.0011) and depression level (A: Rs 0.45, *p* = 0.0004, B: Rs 0.56, *p* = 0.0000) in both study periods, indicating that the more intense the patient’s pain level, the greater the dissatisfaction with their appearance and the higher the level of depression.

Multiple regression was performed to identify factors significantly affecting patients’ level of functioning in basic life activities and mood. In addition, the analysis considered the factors with which significant interdependencies had been previously demonstrated.

Multivariate regression results showed that the model for ADL proved significant (*p* = 0.0131), and all predictors together explained 34% of the dependent variable. In addition, burn area (TBSA), pain level (quantitative measure of pain experienced during the last week—questionnaire part B and measure of pain experienced at the time of the survey-questionnaire part C) and gender had a significant effect on the level of daily functioning (Table 3).

Increasing the TBSA score by one level reduced ADL performance by an average of 0.05 ± 0.02 points. ADL performance in men was, on average, 0.94 ± 0.45 points higher than in women. Greater pain intensity worsened ADL functioning. The subsequent regression analysis was conducted to estimate the effect of selected variables on depression. The model was significant (F = 5.07; *p* = 0.0001), and all predictors explained 46% of the dependent variable together. The level of pain experienced (quantitative measure of pain experienced during the last week—questionnaire part B) significantly affected the level of depression (Table 4).

Increasing the SF-MPQ scale score by one level decreased the BDI scale score by an average of 0.15 ± 0.07 points.

## 6. Discussion

Self-perception takes place on many levels. Among other things, it depends on the individual’s assessment of appearance and satisfaction with it. Dissatisfaction occurs when there is a discrepancy between the expected (ideal) and actual appearance [23,27]. Visible burn scars can be considered the leading cause of dissatisfaction or lack of acceptance of one’s image [27]. Burn scars are not only an aesthetic problem. Various negative symptoms, such as pain or itching, accompany them [28,29]. They can also cause functional incapacity and problems in psychosocial activities.

A consequence of the healing process of burn wounds may be the formation of hypertrophic scars, which are less flexible and worsen the appearance of the person [30]. Hence, it becomes essential to ‘manage’ burn scars to minimise their extent and tendency to overgrow [28,31]. Risk factors for their occurrence include gender, age, location, skin type, wound healing time, type of graft used, the burn severity, multiple surgical procedures, bacterial colonisation and whether the skin has been stretched in the past. In addition, a more extended hospital stay may be a factor in worsening the appearance and healing of the wound [32]. In our study, longer stays correlated with worsening independence in basic activities of daily living and levels of depression.

Hospitalisation time is a difficult time for people with burn wounds. When the lesions occur over a larger body area, including the face or hands, such a location makes it challenging to perform self-care activities. Patients additionally struggle with emotions due to family separation, a long break from work, and the need to adjust to the rhythm of work in the ward [33]. Analysis of patients’ functioning in basic life activities during the hospitalisation period revealed that the burn area’s extent mainly influences patients’ self-efficacy, gender and pain intensity. Extreme individual variability with changing patterns over time characterises pain resulting from burns [34]. Pain complaints affect a person’s functional performance, not only during hospitalisation but also at a later stage [35]. This relationship was pointed out by, among others, Ullrich et al. [35]. They saw that more severe pain one year after the end of treatment was a predictor of lower physical functioning two years after hospitalisation. They found a similar relationship with depression. Considerable depression in the first month after leaving the hospital influenced poorer physical functioning one year after hospitalisation. Our results also showed a significant relationship between pain and the severity of depressive symptoms in the two time periods studied. Patients with higher pain severity showed more severe depressive symptoms than those experiencing lower pain. Esfahlan et al. [36] demonstrated a relationship between psychological and affective reactions and rest pain. In addition, they noted a correlation between mild and acute stress reactions and pain intensity.

In conclusion, this study highlighted that psychological and affective reactions to pain, such as anxiety, anorexia, fear, fatigue, helplessness, hopelessness and depression, cause increased pain. Furthermore, any situation that exacerbates one of these reactions results in increased pain. Therefore, by analysing emotional factors, it can be concluded that they can influence pain experience and behaviour [36]. Furthermore, authors of other studies point to anxiety, depression and pain association in burn patients [34].

In the available literature, the most common predictors of pain were age [37,38] and burned body surface area [38]. However, there are also reports that, after accounting for age and gender, burns had a minor effect out of all types of injury on pain intensity [39]. Our study found no difference in pain intensity scores after accounting for demographic factors other than gender. Women reported a higher intensity of pain.

Our study showed no significant correlation between satisfaction with appearance and depression. Perhaps there was a measurement in too short a time since the injury. Different results were obtained by other authors [25]. They showed that changes in body structure or function as a result of the forming scar lowered patients’ self-esteem and promoted the occurrence of depression [25,40]. In addition, Al Ghriwati et al. [23] noted that at all periods studied, i.e., at hospital discharge, at 6,12 and 24 months after the end of hospitalisation, satisfaction with appearance was significantly correlated with depressive symptoms occurring five years after the burn. Mainly six months after hospital discharge, patients showing higher levels of dissatisfaction with their appearance were characterised by more severe depressive symptoms. The results of our and other authors’ studies emphasise the importance of implementing screening for depression and dissatisfaction with the appearance of burn patients during hospitalisation [23]. Both variables have a subjective dimension and are influenced by various factors, including the level of perceived social support or the sociocultural environment in which the person with burns lives [25]. Caltran et al. [25] showed that patients whose scars were in visible areas reported greater dissatisfaction with their appearance than those with less visible burn marks. In our study, patients indicated one particular scar they felt was the worst possible. As many as 40% of patients indicated a scar located on the hands. Other areas mentioned as either visible or difficult to cover, e.g., in summer, were the forearm, shoulder (21%), chest, face, neck (16%) and lower leg, thigh (23%). The result may have been influenced by the fact that the scars, taking into account the healing process, were not yet mature and were, therefore, very visible.

Moreover, the hands are the part of the body that we look at often during daily activities and whose efficiency affects the level of independence. Throughout their hospitalisation, burn patients struggle with the pain associated with the burn wound and with dressing changes, surgery, rehabilitation and the newly formed scar. Therefore, in addition to monitoring pain levels and the wound healing process, more time should be spent talking, counselling and providing emotional support. The cooperation and education of the family is also an essential element.

## 7. Conclusions

Pain complaints are a significant factor influencing functional status and levels of depression in patients with a burn wound. Dissatisfaction with one’s appearance promotes the development of depressive symptoms. Early recognition of worsening pain symptoms can prevent or reduce symptoms of depression. During hospitalisation, a plan for scar treatment and rehabilitation after hospitalisation should be discussed with the patient. Such measures increase patients’ awareness of wound healing and therapy and promote the strengthening of self-care. In addition, patients receive a positive signal that they can always count on professional help during hospitalisation. They also know what help they need and where they can access it after hospitalisation. If depressive symptoms are identified, the patient should be given appropriate support as soon as possible.

## 8. Limitations

Reports on the post-hospitalisation period preoccupy the available literature. In our study, we referred to assessing the patient’s functioning during hospitalisation. The hospitalised patient faces completely different problems during hospitalisation than those who have completed their treatment and have been discharged home. Therefore, future research should consider patient functioning during and after hospitalisation. Such a research model will make it possible to determine whether the studied variables change during hospitalisation and how they affect the patient’s life after leaving the hospital. A final limitation was that the number of people included in the study needed to be more significant. Undoubtedly, the small burn unit (11 beds) where the study was performed limited access to a more significant number of patients. In addition, the inclusion and exclusion criteria disqualified a large group of patients.

## Figures and Tables

**Figure 1 ijerph-20-01951-f001:**
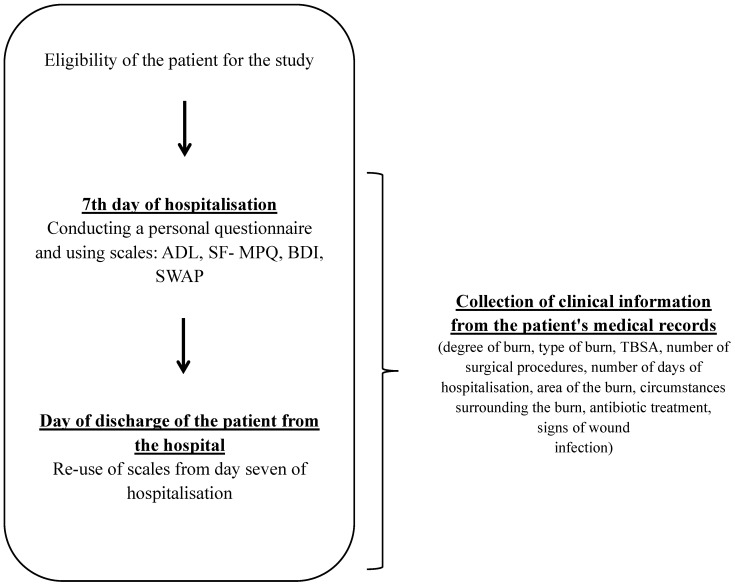
Course of the study.

**Table 1 ijerph-20-01951-t001:** General characteristics of the subjects in terms of sociodemographic and clinical variables.

Variable	Description of the Variable	N	%
**Gender**	Women	17	29.8
	Men	40	70.2
	Total	57	100
**Education**	Comprehensive	4	7.0
	Vocational	28	49.1
	Secondary	18	31.6
	Higher	7	12.3
**Marital status**	Married	36	63.1
	Single	12	21.1
	Divorcee	9	15.8
**Place of residence**	City	37	64.9
	Rural area	20	35.1
**Professional status**	Working	42	73.7
	Pension/allowance	2	3.5
	Retirement	9	15.8
	Unemployed	4	7.0
**Degree of burn**	Partial-thickness	15	26.3
	Partial-thickness/full-thickness	24	42.1
	Full-thickness	18	31.6
**Type of burn**	Thermal	53	92.9
	Electric	1	1.8
	Chemical	3	5.3
**Scar location**	Chest, face, neck	9	15.8
	Lower leg, thigh	13	22.8
	Forearm, arm	12	21.0
	Hands	23	40.4

N—number. min.—minimum. max.—maximum. SD—standard deviation.

**Table 2 ijerph-20-01951-t002:** Comparison of functioning and subjects on day seven of hospitalisation and the day of discharge from the hospital.

Scales	Measurement on Day Seven			Measurement on the Day of Discharge from the Hospital			*p*-Value *
	Average	SD	Median	Average	SD	Median	
**Level of daily functioning (ADL)**	4.39	1.40	4.00	5.84	0.75	6.00	<0.0001
**Pain level according to the SF-MPQ scale, part A**	17.51	8.36	18.00	6.02	4.94	5.00	<0.0001
**Pain level according to the SF-MPQ scale, part B**	64.11	15.18	68.00	27.25	12.82	27.00	<0.0001
**Pain level according to the SF-MPQ scale, part C**	1.49	0.68	1.00	0.84	0.77	1.00	<0.0001
**Depression level (BDI)**	10.91	6.58	12.00	6.46	6.43	4.00	<0.0001
**Level of satisfaction with appearance (SWAP)**	31.47	15.77	33.00	26.46	15.58	24.00	0.0239

* Wilcoxon test. SD—standard deviation.

**Table 3 ijerph-20-01951-t003:** Factors influencing daily functioning (ADL scale)—multiple regression results.

Variables	B	SE	*p*
**Free expression**	4.98	1.21	0.0001
Burned body surface area (TBSA)	−0.05	0.02	0.0126
Number of days of hospitalisation	−0.02	0.02	0.4558
Pain level according to the SF-MPQ scale, part A	−0.02	0.03	0.6437
Pain level according to the SF-MPQ scale, part B	0.04	0.01	0.0004
Pain level according to the SF-MPQ scale, part C	0.40	0.17	0.0208
Level of satisfaction with appearance (SWAP)	−0.01	0.01	0.7132
Depression level (BDI)	0.00	0.03	0.9863
Gender (0—F; 1—M)	0.94	0.45	0.0413
Place of residence (0—rural; 1—city)	−0.77	0.39	0.0532
**F**	2.69; *p* = 0.0131		
**R²**	0.34		

R^2^—coefficient of determination; F—test statistic F.

**Table 4 ijerph-20-01951-t004:** Factors influencing the level of depressive symptoms (BDI scale) in the subjects—multiple regression results.

Variables	B	SE	*p*-Value
**Free expression**	−4.68	5.65	0.4124
Pain level according to the SF-MPQ scale, part A	0.08	0.13	0.5110
Pain level according to the SF-MPQ scale, part B	0.15	0.07	0.0352
Pain level according to the SF-MPQ scale, part C	−0.45	1.26	0.7222
Level of daily functioning (ADL)	−0.09	0.56	0.8674
Level of satisfaction with appearance (SWAP)	0.08	0.06	0.1922
Number of days of hospitalisation	0.18	0.10	0.0673
Gender (0—F; 1—M)	−2.64	1.84	0.1582
Symptoms of wound infection 0—no; 1—yes 0	−0.18	1.86	0.9233
**F**	5.07; *p* = 0.0001		
**R²**	0.46		

R^2^—coefficient of determination; F—test statistic F.

## Data Availability

Data are available from the corresponding author upon reasonable request.

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
