# Peer review of "Wound Pain as a Determinant of Function in Patients Hospitalised for Burns"

_ijerph, 2023, doi:10.3390/ijerph20031951_

Round 1

Reviewer 1 Report

Dear Sir or madame,

thank you for the kind invitation to review the article "Factors influencing the level of functioning in patients hospitalised with burns using multiple regression analysis."

The article aims to assess and analyze the level of functioning in patients hospitalised for burns. Therefore, 57 patients were included in the study. Complex multiple regressions were calculated for the statistical analysis. The results, namely that pain, degree of burns and patient assessment of the scar are important predictors for the psychophysical assessment of functioning, can provide additional knowledge for practitioners to treat these patients better. I appreciate your precious work. The article is interesting and well written.

We would like to add the following points:

1.) Were parameters such as sepsis, multi-organ failure or a prognostic score such as the ABSI score recorded? If so, do they play a role in the regression analysis?

2.) For the surgical readership, a brief description of surgical therapy would be advisable. Did the surgical therapy (for example tangential vs. epifascial necrosectomy) made any difference for the statistical analysis and patient assessment of the scars?

Thank you very much.

Best Regards

Author Response

Dear Reviewer,

We sincerely appreciate your review. We would like to add the following points:

  1. The aforementioned parameters were not recorded, the study was conducted in a burn ward and not in an intensive care ward.
  2. The study did not take into account the type of surgical treatment, only the number of surgical procedures.

Thank you very much.

Best regards

Reviewer 2 Report

Manuscript Number: IJERPH-2039220

Title: Factors influencing the level of functioning in patients hospitalised with burns using multiple regression analysis

Thank you to the authors for presenting their work. Overall the elements of a good study report or reports are evident. However, the study is confused and requires deeply considered adjustments to the methods. The report requires significant focus, to provide the reader with adequate justification to accept the findings and further, to consider a change in practice in response.

While potentially novel, the inpatient context of the study raises significant questions as to the validity of the scales used. Thus, the aims of the study may also need to be adjusted to align with the interpretation of the outcomes as able in the context of the patient journey.

Perhaps the best way to address the issues is to split this report into multiple manuscripts that allow the authors to specifically justify and focus on different outcomes in each separate piece of writing?

Abstract – Too long. Please shorten and adjust after responding to the following:

Intro

1.            The Introduction is convoluted and confusing. The justification of the study is lost in the prose. Please focus the Introduction on justification of the study as proposed which seems to be primarily about the factors influencing function.

2.            The expression in the Introduction requires adjustment. Pain is not a function, yet it is bundled into the core premise of the study.

Methods

3.            The authors imply that the BDI is a diagnostic tool for depression. If that is the case, then the validity of the BDI to diagnose depression during an inpatient stay, particularly when the maximum duration was 55 days, must be justified and confirmed. How can a reader accept that participants had diagnosable depression during their inpatient stay when there are so many acute factors which would confound such a diagnosis?

4.            What is the minimum clinically important difference for the BDI scale? While there is a statistical difference, are the results meaningful for a clinician?

5.            Is the SWAP a valid survey during an acute inpatient stay? Would a patient have time to develop an understanding of their appearance and how it might impact their daily life without experiencing their life outside the hospital?

6.            The validity of use of the POSAS in the timeframes of the study, requires justification. At discharge, is there actually a scar to assess? This reviewer suggests that there isn’t adequate validation to model it as an outcome using the available variables.

7.            The statistical modeling seems to be a fishing expedition and requires further explanation and focus. The methods consequently require adjustment. With only 57 participants, and the methods described, the likelihood of overfitting with the multitude of covariates of interest is high, affecting the ability of the reader to interpret the results.

8.            As 1/3 of the participants had signs of infection for their whole hospital stay, was this included as a covariate in the statistical modeling? If not, why not - as infection is a significant factor influencing pain and scar?

9.            Characteristics of the Studied Group – Is unnecessarily long because it presents data more than once. Many of the descriptives are duplicated in the text and the table. Please reconcile and shorten this section. TBSA should be in the Table as it is a key variable in all burns studies.

10.          Please contemporize the presentation of burn depth. No longer is burn depth described by numbers, rather descriptive categorization should be used.

Results 

11.          Adjust after reconsidering the Methods. Only include that which pertains to the study report focus. If ADL’s is to remain the main focus, drop the models for all other outcomes, present appropriate descriptives of variables that impact function and use those variables (only) as covariates.

12.          Depression and scarring are quite separate themes and study reports with appropriate Introduction and Methods in line with those themes would be much easier to digest.

Discussion

13.          Confusing and meanders around the main Present the main finding of the study first and then work the Discussion in the same order as outcomes are presented in Methods and then Results.  

Conclusion

14.       Adjust after addressing above comments.

Author Response

Dear Reviewer,

Thank you very much for your review. Following your valuable suggestions, we made significant improvements to make the article more understandable in interpreting the results and clarifying the purpose of the study. The article has been substantially revised and modified in line with the comments received. We tried to adapt to all comments, which is why we resigned from some results. We have changed the title of the article and adapted the content of the introduction to the new title. We have corrected the abstract according to the topic of the article. Furthermore, we had the text translated by a professional translator.

Methods

Answer to 3 and 4 together

We used the verified Beck Depression BDI scale, used both in clinical and research settings.

Below we present two articles that contain studies conducted in clinical conditions with the BDI scale.

Loncar, Z.; Bras, M.; Micković, V. The Relationships between burn pain, anxiety and depression. Coll Antropol. 2006,30(2),319-325.

Arif M, Ramprasad KS. Prevalence of anxiety and depression in burns patients in a Tertiary Care Hospital. IOSR-JDMS. 2013;10(4):6-9.

We used the key to interpret the results prepared by the authors:

The following are the ranges of depression: 0-11 points - no depression; 12-26 points - mild depression; 27-49 points - moderately severe depression; 50-63 points - very severe depression.

Hajduk, A.; Korzonek, M.; Przybycień, K.; Ertmański, S.; Stolarek, J. Study of depressiveness with Beck Depression Inventory in patients with cardiac arrhythmias. Annales Academiae Medicae Stetinensis 2011,57(1),45-48.

  1. We are aware that the scale is screening. We were more interested in showing that a burn as a traumatic event significantly lowers the mood and, in the absence of proper support, may later lead to the development of depression.

In our study, the use of the SWAP scale was of great importance. By using it twice (on the 7th day and on the day of discharge from the hospital), the patients assessed themselves (their bodies) when they had burn wounds, and then scars. This was primarily to focus the patient's attention on what he/she thinks and feels when he/she sees himself/herself. Using the SWAP scale during hospitalization, medical staff (including a psychologist) can prepare the patient for how his appearance will change and outline the problems he/she may encounter after leaving the hospital. Similarly to our study, the authors of other studies also used the SWAP scale on the day of discharge from the hospital.

McAleavey AA, Wyka K, Peskin M, Difede J. Physical, functional, and psychosocial recovery from a burn injury are related and their relationship changes over time: A Burn Model System study. Burns. 2018;44(4):793-799.

Al Ghriwati N, Sutter M, Pierce BS, Perrin PB, Wiechman SA, Schneider JC. Two-year gender differences in Satisfaction With Appearance after burn injury and prediction of five-year depression: A latent growth curve approach. Arch Phys Med Rehabil. 2017;98(11):2274-2279.

  1. The use of the POSAS scale was justified, but in order to maintain the clarity of the article, as suggested by the reviewer, we resigned from it in the following study. We have included the scales twice: on the 7th and the last day in the hospital.
  2. We have changed the statistical description and the number of regressions.
  3. We have included information about infections in the text. It was an important element when monitoring the health of patients as well as collecting and analyzing data.

Answer to 9 and 10 together:

We have changed the description of the group a bit. We were not able to include TBSA in the table because it would spoil its structure. TBSA is described below the table. A descriptive categorization was used for the depth of burns.

Results

Answer to 11 and 12 together:

We made the most substantial modification in the description of the results. They ought to be more understandable now.

Discussion and conclusion

Answer to 13 and 14 together:

The discussion has been revised according to the results. The conclusions have been corrected.